# *De Novo* Assembly and Comparative Analysis of the Complete Mitochondrial Genome of *Chaenomeles speciosa* (Sweet) Nakai Revealed the Existence of Two Structural Isomers

**DOI:** 10.3390/genes14020526

**Published:** 2023-02-19

**Authors:** Pei Cao, Yuan Huang, Mei Zong, Zilong Xu

**Affiliations:** 1Institute of Sericulture and Tea, Zhejiang Academy of Agricultural Sciences, Hangzhou 310021, China; 2College of Life Sciences, Anqing Normal University, Anqing 246133, China

**Keywords:** *Chaenomeles speciosa*, mitochondrial genome, chloroplast, IGT, MTPTs, homologous recombination

## Abstract

As a valuable Chinese traditional medicinal species, *Chaenomeles speciosa* (Sweet) Nakai (*C. speciosa*) is a natural resource with significant economic and ornamental value. However, its genetic information is not well understood. In this study, the complete mitochondrial genome of *C. speciosa* was assembled and characterized to explore the repeat sequences, recombination events, rearrangements, and IGT, to predict RNA editing sites, and to clarify the phylogenetic and evolutionary relationship. The *C. speciosa* mitochondrial genome was found to have two circular chromosomes as its major conformation, with a total length of 436,464 bp and 45.2% GC content. The mitochondrial genome contained 54 genes, including 33 unique protein-coding genes, 18 tRNAs, and 3 rRNA genes. Seven pairs of repeat sequences involving recombination events were analyzed. Both the repeat pairs, R1 and R2, played significant roles in mediating the major and minor conformations. In total, 18 MTPTs were identified, 6 of which were complete tRNA genes. There were 454 RNA editing sites in the 33 protein-coding sequences predicted by the PREPACT3 program. A phylogenetic analysis based on 22 species of mitochondrial genomes was constructed and indicated highly conserved PCG sequences. Synteny analyses showed extensive genomic rearrangements in the mitochondrial genome of *C. speciosa* and closely related species. This work is the first to report the *C. speciosa* mitochondrial genome, which is of great significance for conducting additional genetic studies on this organism.

## 1. Introduction

*C. speciosa*, belonging to the genus *Chaenomeles* in the Rosaceae family, is a native temperate plant widely cultivated in Asia and Europe. *C. speciosa* has been widely used in medicine and the functional food industries [1,2]. The dried fruit of *C. speciosa* has been utilized in traditional Chinese medicine for thousands of years. Previous studies have shown that the chemical components of *C. speciosa* extracts are rich in flavonoids, phenolics, terpenoids, and phenylpropionic acids [3,4,5]. The People’s Republic of China’s Pharmacopoeia lists that oleanolic acid and ursolic acid are two triterpenoid acids that can be used in medicine. Several traditional *C. speciosa* cultivars have recently been released on the market and undergone increased production [6].

Mitochondria are involved in numerous metabolic processes and perform crucial roles in cell differentiation, apoptosis, cell development, and cell division by converting biomass energy into chemical energy for daily activities [7,8,9]. On the other hand, the mitochondrion is a semi-autonomous organelle with a mitochondrial genome genetic system distinct from the cell nucleus and typically exhibits maternal inheritance [10,11]. Coordinated nucleocytoplasmic interaction is essential for plant growth. Artificial hybridization between or within plant species could change the balance of cytonuclear interactions, often used to breed new varieties [12]. Cytoplasmic male sterility (CMS), driven by a genetic conflict between the nuclear and mitochondrial genomes, is a typical example and has been reported in many species [13].

Plant mitochondria contain the largest and most complex organelle genomes, with various sizes and structures [14,15]. These mitochondria are recognized as obtaining foreign sequences through intracellular gene transfer (IGT) and horizontal gene transfer (HGT) [16,17,18,19]. Numerous well-known genes, introns, and varied intergenic regions can be found in mitochondrial genomes [20]. The size of mitochondrial genomes varies significantly, by up to three orders of magnitude, because of differences in non-coding regions, such as repeat sequences, introns, intracellularly transferred sequences from plastids (mitochondrial plastid sequences, MTPTs), and horizontal gene transfer from foreign donors [21]. As previously reported, the *Viscum scurruloideum* mitochondrial genome exhibited a size of 66 Kb [22], while the *Lactuca sativa* and *Siberian larch* mitochondrial genomes were estimated at 11.7 Mb [23,24]. The mitochondrial genome was previously thought to have a conventional, single-circular chromosome structure. However, previous reports indicate that the conformations are diverse, with polycyclic being the most common in *Cucumis sativus* [25] and a variety of linear, circular, and branched structures in *Quercus acutissima* [26]. Moreover, the multi-chromosomal mitochondrial genome has been reported in many species. For example, 54 distinct circular chromosomes in *Lophophytum mirabile* range from 7.2 to 580 kb in size [27], while 21 smaller circular chromosomes in *Rhopalocnemis phalloides* range from 4.95 to 7.86 kb [28].

Plant mitochondrial genomes have a dynamic structure with multiple configurations [29,30]. It has generally been accepted that large repeats of several kilobases of DNA mediate the presence of multiple mitochondrial genome conformations via recombination at a high frequency [31]. Emily et al. [32] found that only 13 of the 72 species of angiosperms showed no repeats with lengths exceeding 600 bp, and the repeats were more extensive and more frequent in vascular plants. Further, the accumulation of sub-genomic interconvertible through recombination events could change the flexibility of plant mtDNA and help mtDNA evolve through genetic diversity [14,31,33]. The abundance of repeat sequences is typically related to mitochondrial genome structural rearrangements [34]. In addition to the mitochondrial genome’s variability in genome size, gene content, and genomic structure, functional genes also differ significantly due to post-transcriptional editing, namely RNA editing, which may lead to incredibly diverse gene sequences [35,36]. Despite the high rate of rearrangement and recombination, the mitochondrial genome typically shows a low mutation rate, which may result from the mitochondrial repair processes [14,33,37].

Understanding genetic information is of great significance, but, until now, only the chloroplast genome of *C. speciosa* has been sequenced and studied [38]. In this study, the mitochondrial genome of *C. speciosa* was first assembled and annotated. This study aimed to analyze the mitochondrial genome structure mediated by repeated sequences, IGTs, RNA editing sites, and selective pressure and explore the synteny and phylogenetic relationships. Our results expand the genetic information in Rosaceae and provide a theoretical basis for further utilization and genomic breeding studies of the *C. speciosa* species.

## 2. Materials and Methods

### 2.1. Plant Materials, DNA Extraction, and Sequencing

The fresh *C. speciosa* materials were collected from Houyi Garden of Southwest University, 2 Tiansheng Road, Beibei District, Chongqing (longitude 106.4242178°, latitude 29.8251892°, or 106°25’ 27.184” E, 29°49’ 30.681” N). Total genomic DNA was obtained using the cetyltrimethylammonium bromide (CTAB) method [39]. The DNA library (Illumina) was constructed with an insert size of 500 bp using the NEBNext^®®^ Ultra™ II DNA Library Prep Kit (NEB, Beijing, China) [40]. The Hiseq X platform was used for short-read sequencing (Novogene, Tianjin, China) [41]. Purified DNA was prepared for long-read sequencing with the SQK-LSK108 genomic sequencing kit (Oxford, UK). The purified library was loaded into an R9.4 Spot-On Flow for nanopore sequencing.

### 2.2. Assembly and Annotation of the Mitochondrial Genome and Chloroplast Genome

The organelle genome in this study was assembled using a hybrid assembly strategy. First, GetOrganelle (version v1.7.5) [42] was used to obtain short reads of the chloroplast and mitochondrial genomes separately. The short reads were then assembled by SPADES [43] into a unitig graph. The contigs in the unitig graph served as the reference genomes for the nanopore read mapping using BWA software [44]. The mapped reads were extracted using Samtools (v1.9) [45], and then the long reads mapped to the repeat region of the unitig graph were used for solving the repeat region. Finally, the contigs were merged using bandage software [46] to form the circular DNA molecule. The mitochondrial genome was annotated by the Geseq software (v2.03) [47]. CPGAVAS2 (v1.0) [48] was utilized to annotate the chloroplast genome with Database 3. The chloroplast genome annotation was checked via the CPGView web server (v1.0) [49].

### 2.3. Analysis of Repeated Sequences and Genome Recombination

BLASTN [50] was used to detect the repeated sequences in the *C. speciosa* mitochondrial genome, followed by manual exclusion. Then, two repeated units of the repeated sequences and their flanking 1000 bp regions were extracted as the primary conformation. Afterward, the repeat unit’s 1000 bp regions upstream and downstream were exchanged to simulate the secondary conformation that could result from artificial recombination. BWA software mapped the nanopore data to major and minor conformation sequences [44]. By counting the number of long reads that completely overlapped the regions of the repeated sequence, it was feasible to determine whether there was mitochondrial genome recombination.

### 2.4. DNA Transfer and Phylogenetic Tree Construction

Due to the lack of a published nuclear genome for *C. speciosa*, only two organelle genomes can be utilized to analyze intracellular sequence migration. To identify the MTPTs, the chloroplast genome of *C. speciosa* was compared with the mitochondrial genome using the BLASTN program with the parameter “-evalue 1e-5”. The results were visualized using TBtools (v1.112) [51].

Mitochondrial genome data of 21 species from 6 families of angiosperm were selected and downloaded. PhyloSuite (v1.1.16) was used to extract the common protein-coding genes (PCGs) [52]. The MAFFT in PhyloSuite was used for multiple sequence alignment [53]. A maximum likelihood (ML) phylogenetic tree was constructed using IQtree (v1.6.12) (bootstrap = 1000) [54]. GTR + F + R2 was chosen as the best-fit model according to the Bayesian information criterion (BIC) [55]. Bayesian inference (BI) analysis was performed using MrBayes (v3.26) with default parameters and 1000 bootstrap replicates [56]. ITOL software (v6) was used to visualize the results of the phylogenetic analysis [57].

### 2.5. RNA Editing Site Identification and Synteny Analysis

The PCGs of *C. speciosa* were extracted using PhyloSuite software. The extracted PCGs were submitted to the PREPACT3 web server (http://www.prepact.de/prepact-main.php, accessed on 3 August 2022) for RNA editing site prediction with the default parameters [58].

The mitochondrial genome data of seven species (*Pyrus betulifolia*, *Sorbus torminalis*, *Malus domestica*, *Eriobotrya japonica*, *Prunus avium*, *Rosa chinensis*, and *Fragaria orientalis*) from the Rosaceae family that properly represent their genera were downloaded from the NCBI public database at https://www.ncbi.nlm.nih.gov (accessed on 3 August 2022). BLASTN was performed to compare eight mitochondrial genomes pairwise and obtain homologous sequences. For the Multiple Synteny Plot, conserved colinear blocks longer than 0.5 Kb were selected. The Multiple Synteny Plot of *C. speciosa* with the seven species was constructed based on sequence similarity using MCscanX [59].

### 2.6. dN/dS Analysis

Twenty-eight genes from 12 Rosaceae species’ mitochondrial genomes were series connected for polygenic nucleotide alignment to estimate the selective pressure at the species level. The 12 species were *P. betulifolia*, *Pyrus pyrifolia*, *M. domestica*, *Malus hupehensis*, *Sorbus aucuparia*, *S. torminalis*, *C. speciosa*, *E. japonica*, *P. avium*, *Prunus salicina*, *F. orientalis*, and *R. chinensis*. The Yn00 module in PAML (v4.9) [60] was used to estimate the separate non-synonymous substitutions (dNs), synonymous substitutions (dSs), and dN/dS rates. The parameters were as follows: verbose = 0, icode = 0, weighting = 0, common 3×4 = 0 (use one set of codon frequencies for all pairs), ndata = 1. The boxplot and heatmap were created using the R-package (v3.2.2) (ggplot2 and heatmaply) [61,62].

## 3. Results

### 3.1. General Features of the C. speciosa Mitochondrial Genome

The major conformation of the *C. speciosa* mitochondrial genome was two circular chromosomes, with a total length of 436,464 bp and 45.2% GC content (Figure 1). The lengths of the two circular chromosomes (chromosome 1 and chromosome 2) were 307,720 bp and 128,744 bp, and the GC contents were 45.15% and 45.33%, respectively (Table 1).

The *C. speciosa* mitochondrial genome was annotated with 33 unique PCGs, including 24 mitochondrial core genes, 9 variable genes, 18 tRNA genes (2 were multiple copies), and 3 rRNA genes (Table 2). The set of core genes included five ATP synthase genes (*atp*1, *atp*4, *atp*6, *atp*8*,* and *atp*9), nine NADH dehydrogenase genes (*nad*1, *nad*2, *nad*3, *nad*4, *nad*4L, *nad*5, *nad*6, *nad*7, and *nad*9), four cytochrome C biogenesis genes (*ccm*B, *ccm*C, *ccm*Fc, and *ccm*Fn), three cytochrome C oxidase genes (*cox*1, *cox*2, and *cox*3), one transport membrane protein gene (*mtt*B), one maturase (*mat*R), and one cytochrome c biogenesis gene (*cob*). The non-core genes consisted of three large subunits of ribosome genes (*rpl*5, *rpl*10, and *rpl*16), five small subunits of ribosome genes (*rps*1, *rps*3, *rps*4, *rps*12, and *rps*13), and one succinate dehydrogenase gene (*sdh*4). Additionally, two tRNAs, *trn*F-GAA and *trn*M-CAU, were represented by two copies.

### 3.2. Repeats and Homologous Recombination

As shown in Figure 2, seven pairs of repeated sequences were detected as being involved in mediating mitochondrial genome recombination, including R1 and R2 as long repeats (>1000 bp), R15 as short repeats (<100 bp), and R4, R5, R7, and R14 as medium-sized repeats. Among them, R4, R7, R14, and R15 were reverse repeats; R1, R2, and R5 were forward repeats. In addition, R2, R7, and R14 were located separately in chromosomes 1 and 2, and the other four pairs of repeated sequences were located in chromosome 1 (Table 3).

After simplifying, four chromosomes were presented (Table 4) and mediated by R1 and R2. As shown in Figure 3, R1 and R2 may mediate mtDNA formation into three independent circular chromosomes or one united circular chromosome. The major conformation was two circular chromosomes, which were supported by the majority of long reads. R1, with the most extensive length, was 7887 bp and obtained similar rates, supporting long reads for both routes from the major conformation (combined in one circular) and minor conformation (divided into two circulars) (21 + 20/16 + 18). In contrast, another pair of R2 repeated sequences allowed for many more reads on a single circular chromosome than on two circular chromosomes (168 + 100/3 + 0).

In addition, as large repeats involving recombination events, R1 exhibited a recombination frequency of 45.33%, while R2 showed a recombination frequency of 1.1%. The other repeats were involved in recombination, with frequencies ranging from 0.22% to 2.27%.

### 3.3. Gene Transfers between Chloroplast and Mitochondrial Genomes

According to the sequence similarity analysis, 18 MTPTs (Appendix A), with a total length of 3237 bp, accounted for 0.74% of the total length of the mitochondrial genome (Figure 4). Among them, five MTPTs exceeded a length of 100 bp, with MTPT11 and MTPT12 being the longest at 879 bp. By annotating these homologous sequences, six complete tRNA genes were identified, including *trn*D-GUC, *trn*H-GUG, *trn*I-CAU, *trn*M-CAU, *trn*N-GUU, and *trn*W-CCA. In addition to two partial tRNA genes (*trn*A-UGC and *trn*A-UGC) and two partial rRNA genes, five unique partial PCGs were identified, including *psbC*, *psbA*, *ndhC*, *atpA*, and *psbE*.

### 3.4. Phylogenetic Analysis

To understand the evolutionary status of the *C. speciosa* mitochondrial genome, a phylogenetic tree of 22 species from six families of angiosperm was constructed (Figure 5). Two mitochondrial PCGs from Brassicaceae (*A. alpina* and *B. oleracea*) were established as an outgroup. The taxa from six families (Rosaceae, Rhamnaceae, Cannabaceae, Moraceae, and Ulmaceae) were well clustered. In the cluster of the Rosaceae family, species from the *Pyrus*, *Malus*, and *Prunus genera* were well grouped.

### 3.5. RNA Editing Sites Prediction

Four hundred fifty-four potential RNA editing sites were predicted on 33 mitochondrial PCGs (Appendix A). The predicted RNA editing sites in each gene are shown in Figure 6. All the RNA editing events ranged from C to U. The *ccm*FN and the *mtt*B genes were edited most frequently, 38 and 34 times, respectively. In total, eight genes (*atp*9, *rps*1, *rps*12, *sdh*4, *atp*8, *rps*3, *rpl*10, and *nad*4L) were edited no more than five times. No potential RNA editing events were predicted in the *nad*4L gene.

### 3.6. Synteny Analysis

The colinear blocks were not arranged in the same order among individual mitochondrial genomes (Figure 7). A significant number of blocks was detected among *C. speciosa*, *M. domestica*, and *E. japonica*. *R. chinensis*, *S. torminalis*, and *P. betulifolia* shared numerous conserved colinear blocks. On the contrary, colinear blocks among *R. chinensis*, *F. orientalis*, and *P. betulifolia* were relatively weak.

### 3.7. Evolutionary Selection Pressure Analysis

As shown in Figure 8, the dN/dS values at the species level by series-connecting 28 PCGs were conducted (Figure 8A). In *C. speciosa*, the dN/dS values were 0.32–1.12, which were higher in genera *Pyrus*, *Malus*, and *Sorbus* than in *Rosa*, *Fragaria*, *Prunus*, and *Eriobotrya*.

The pairwise dN/dS values of all PCGs are shown in Figure 8B. The dN/dS values of PCGs from *R. chinensis*, *F. orientalis*, *P. salicina*, and *P. avium* differed significantly. The dN/dS values of the pairwise comparison from the other eight species (*E. japonica*, *P. betulifolia*, *P. pyrifolia*, *M. domestica*, *M. hupehensis*, *S. aucuparia*, *S. torminalis*, and *C. speciosa*) varied slightly, except for three genes (*cox*2, *mat*R, and *rps*3) from *C. speciosa*.

As shown in Figure 8C*,* the *rps1* gene showed a dN/dS value over 1.0, implying possible positive selection. In comparison, the dN/dS values of the rest of the genes revealed diverse differences among species. N*ad4L* exhibited higher dN/dS values among *R. chinensis* and *F. orientalis* but low dN/dS values among the other eight species. *Atp9*, *cox*1, *cox*3, *nad*5, and *nad*9 showed low dN/dS values among all the species, which indicated they had undergone negative selection during evolution.

## 4. Discussion

### 4.1. Features of the C. speciosa Mitochondrial Genome and IGT Events

Our study produced the first detailed characterization of the complete mitochondrial genome of *C. speciosa*. The size of the *C. speciosa* mitochondrial genome is equivalent to that of *E. japonica* and matches within the range of previously reported mitochondrial genomes in the Rosaceae family. The GC content of the *C. speciosa* mitochondrial genome is 45.20%, equaling the average 45% of the 38 species in the Rosaceae family [63].

Structurally, the two circular chromosomes forming the major conformation of the *C. speciosa* mitochondrial genome demonstrated multi-chromosomes, as previously reported for many plants. The mitochondrial genomes of *C. sativus* and *Q. acutissima* are assembled into three chromosomes [25]. Additionally, the mitochondrial genome of *Silene conica* is arranged into numerous circular chromosomes, though some exhibit no protein-coding capability [23].

Among the 41 protein-coding genes from the mitochondrial genome of the common ancestor of angiosperms, 33 genes were annotated in the *C. speciosa* mitochondrial genome, including 29 core genes and 4 non-core genes. This result indicates that the deleted relevant genes might have been transferred to the nucleus, a common phenomenon during long-term angiosperm evolution [64,65]. Transferring functional genes from the mitochondria to the nucleus is an ongoing process that has helped both the mitochondria and the nucleus change over time [12]. Liu et al. [66] found all 41 genes in Cycads and Ginkgo mitochondrial genomes. In comparison, *Gnetum gnemon* and *Welwitschia mirabilis* lost 11 genes, and 4–7 intact lost mitochondrial genes were found after further searching the transcriptomes, representing the 4–7 mitochondrial genes being transferred to the nucleus. Additionally, in *Gossypium raimondii*, nearly all mitochondrial genes are transferred to the nucleus on Chr1. Some of these genes have more than one copy on different chromosomes [67]. As for other genes, it is possible that they are not activated by acquiring a promoter and other regulatory elements, or they are active with low or specific undetected expression. In contrast to functional gene transfers, nonfunctional gene transfers are also possible. Mitochondrial pseudogenes have been found in the nuclei of a wide range of animal species [68].

IGTs also occurred between the chloroplast, plastid, and nucleus. In the *C. speciosa* mitochondrial genome, MTPTs 1–10 were located on chromosome 1, MTPTs 11–18 were located on chromosome 2, and no MTPTs were found in the repeated sequences. Six complete tRNAs were found in the mitochondrial genome that migrated from the chloroplast, which similarly occurs in other plants’ mitochondria. In both *Suaeda glauca* [69] and *G. raimondii* [67], it was shown that 8 and 15 tRNAs were transferred from the chloroplast genome to the mitochondrial genome. However, protein-coding genes were less numerous and frequent.

### 4.2. The Conformations of C. speciosa Mitochondrial Genome Mediated by Repeated Sequences

The mitochondrial genome is likely composed of several subgenomic chromosomes produced by reversible recombination events, mediated by repeated sequences [30]. In contrast to the circular molecule previously described, the mitochondrial genome structure has appeared as a variable and dynamic combination in many species [70]. As previously reported, repeated sequences play an essential role in shaping the mitochondrial genome structures through genome rearrangements, recombination, and duplications [71]. Large repeats (>1000 bp) may induce mitochondrial genome isomerization during more frequent recombination activity, whereas short- (100 bp) and medium-sized repeated sequences (100–1000 bp) show minor to moderate recombination activity [72,73,74]. As reported in the Rosaceae family, the recombination frequency of repeated sequences ranges from 0.33% to 89.69%. In the repeats of recombination frequency over 50%, 95% are large repeats (19/20), while in the repeats of recombination frequency below 1%, short- and medium-sized repeated sequences account for 99.56% [63].

In the *C. speciosa* mitochondrial genome, seven pairs of repeated sequences were detected as involved in recombination events. The most extensive repeated sequences of R1 were detected, mediating the conformational changes at a frequency of nearly 50%. This finding indicates that R1 medicates the presence of the major and minor conformations in nearly equal proportions. The situation discovered in the mitochondrial genome of *C. speciosa* was similar to that found in the mitochondrial genome of *Scutellaria tsinyunensis* [73], with one extensive repeated sequence forming the major and minor conformations. Nearly equimolar recombined molecules in the mitochondrial genome were generated by large repeats [75], as found in our study, as well as those reported in *Silene vulgaris* [76] and *Ginkgo biloba* [77].

Except for R1, which has a high recombination frequency, the rest of the repeated sequences took part in recombination events at relatively low rates (less than 3% in the *C. speciosa* mitochondrial genome). Such low recombination frequencies have been reported in other plants. In *P. salicina*, nine pairs of medium-sized repeats were recombined at frequencies ranging from 0.55% to 5.7% [78]. In *Nymphaea colorata*, seven medium-sized repeated sequences exhibited a frequency range of 0.11% to 1.28% [72].

Short- and medium-sized repeats rarely participate in recombination, in contrast to large repeats, which have a high frequency of recombination. This can further lead to complex rearrangements [30], which have been hypothesized to involve Break-Induced Replication (BIR) [79] and Single-Strand Annealing (SSA) pathways [80]. Small repeats are generally assumed to recombine sporadically and irreversibly and produce new and stable DNA arrangements. Furthermore, recombinations of short repeat sequences may contribute significantly to heteroplasmy and the evolution of plant mtDNA [81,82]. In *Arabidopsis thaliana*, medium-sized repeats can recombine in response to genome damage or DNA maintenance mutants [83].

### 4.3. Evolutionary Analysis and dN/dS Analysis

The inconsistent order of the colinear blocks’ arrangement suggested that the *C. speciosa* mitochondrial genome had undergone extensive genomic rearrangements, likely contributing to the evolution and diversification of the *C. speciosa* mitochondrial genome.

Most of the mitochondrial genes underwent neutral and negative selections and were highly conserved. The genes analyzed in the *C. speciosa* mitochondrial genome exhibited mostly neutral and negative selections. However, the *rps*1 gene, with a dN/dS value over 1.0 in most species, suggests that it might have undergone positive selection during the evolution of Rosaceae. The *rps*1 gene is among the most dynamic genes and has been found in rice [84], maize [85], and tobacco [86]. However, in *Medicago sativa*, *rps*1 seemed to be a pseudogene, and a functional copy was located in the nucleus [87], which might provide evidence for the intercellular transfer from the mitochondria to the nucleus.

## 5. Conclusions

In this study, the *C. speciosa* mitochondrial genome was assembled and annotated. The total length of the *C. speciosa* mitochondrial genome was 436,464 bp, with 45.2% GC content, consistent with related species from the Rosaceae family. The *C. speciosa* mitochondrial genome was found to be composed of multiple circular chromosomes. R1 altered the conformation of the *C. speciosa* mitochondrial genome by recombination events with high frequency, whereas R2 achieved it through low frequency. Extensive genomic rearrangements were exhibited in the mitochondrial genome of *C. speciosa* and closely related species, while the protein-coding genes were highly conserved. Our study enriches the genetic information for the genus Rosaceae and provides an essential basis for protecting genetic information and improving the molecular breeding of *C. speciosa*.

## Figures and Tables

**Figure 1 genes-14-00526-f001:**
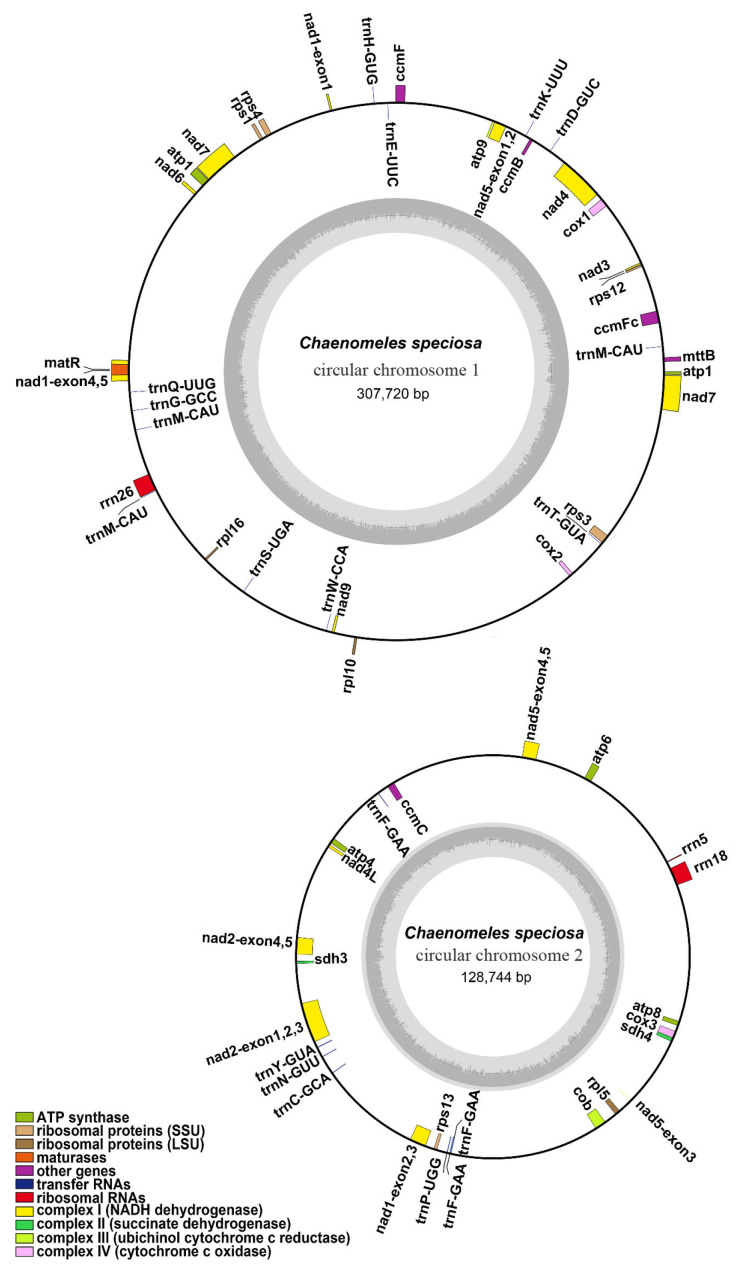
*C. speciosa* mitochondrial genome gene map. For both circular chromosomes, inner genes are transcribed clockwise and outer genes are transcribed counter-clockwise. Genes are color-coded, as indicated in the legend.

**Figure 2 genes-14-00526-f002:**
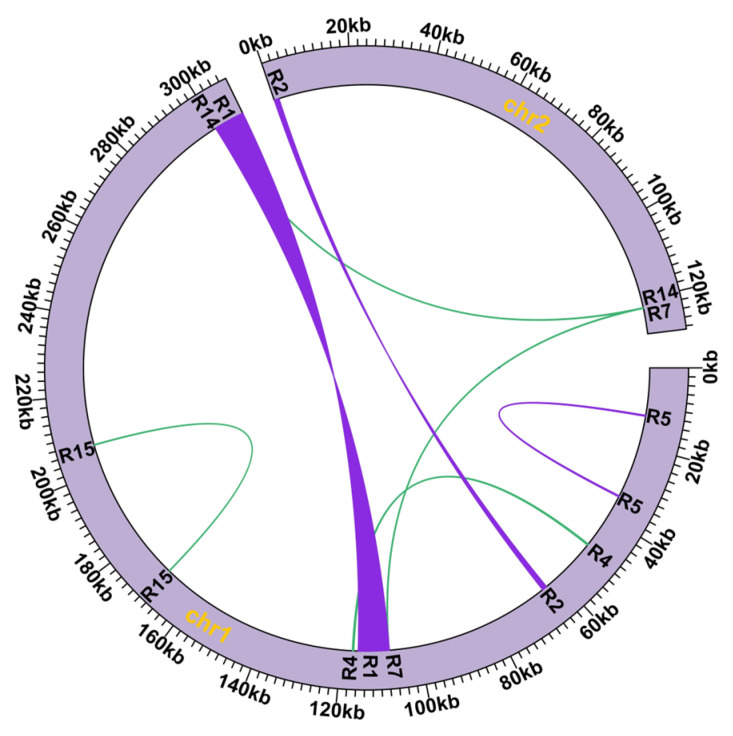
The location of seven repeated sequences in the *C. speciosa* mitochondrial genome. The green and purple lines inside the circle represent seven repeated sequences, and the pairs of R2, R7, and R14 were found on two different chromosomes, while the rest of the repeat pairs were found on chromosome 1.

**Figure 3 genes-14-00526-f003:**
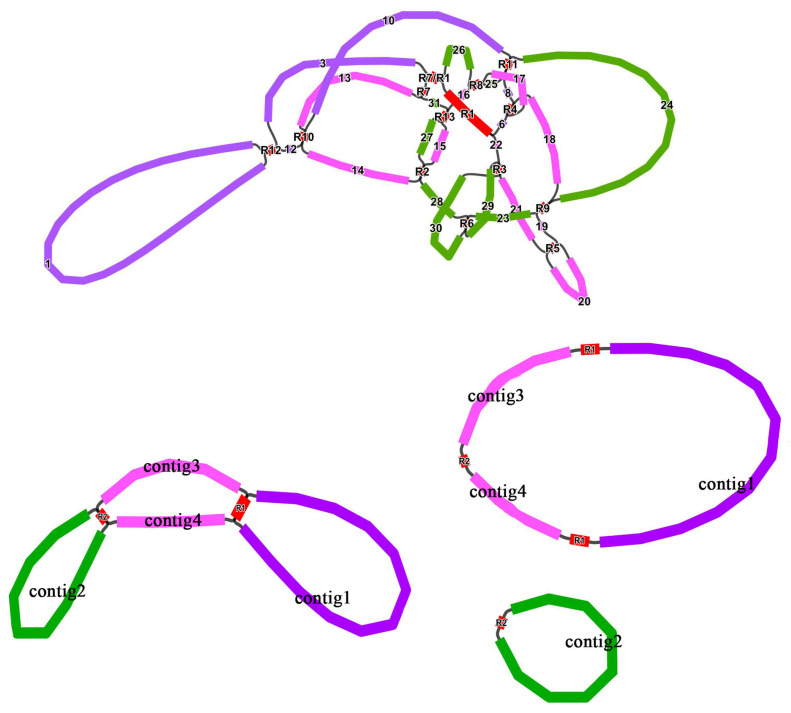
The assembly graph of the *C. speciosa* mitochondrial genome. Colored segments were named contig/R 1–15. Long reads supported all segment adjacencies.

**Figure 4 genes-14-00526-f004:**
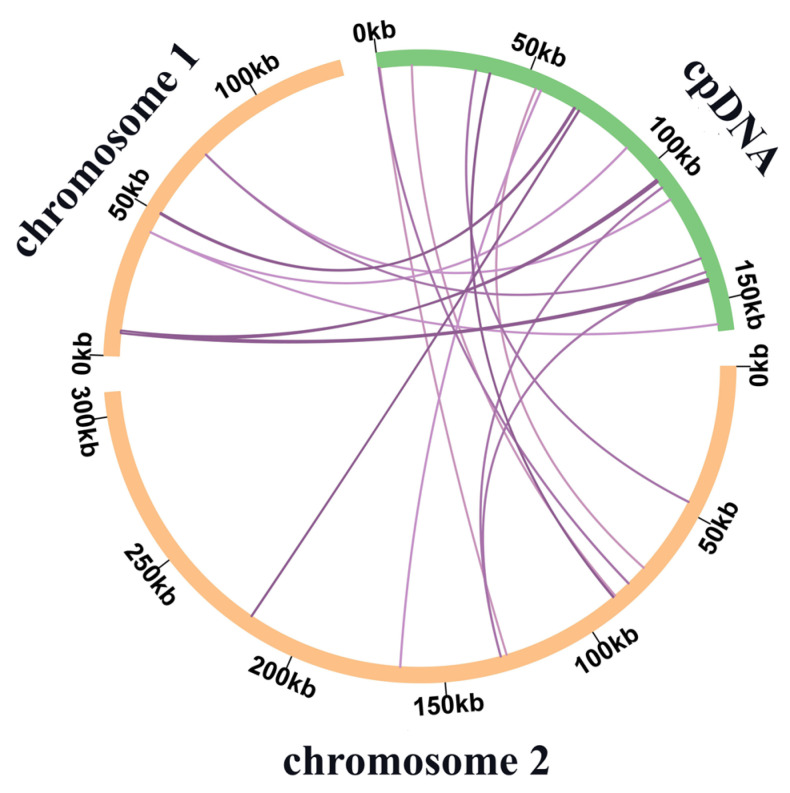
Schematic of 18 MTPTs of *C. speciosa*. MTPTs 1-10 were found in chromosome 1, and MTPTs 11-18 were found in chromosome 2. Appendix A presents the detailed data for 18 MTPTs.

**Figure 5 genes-14-00526-f005:**
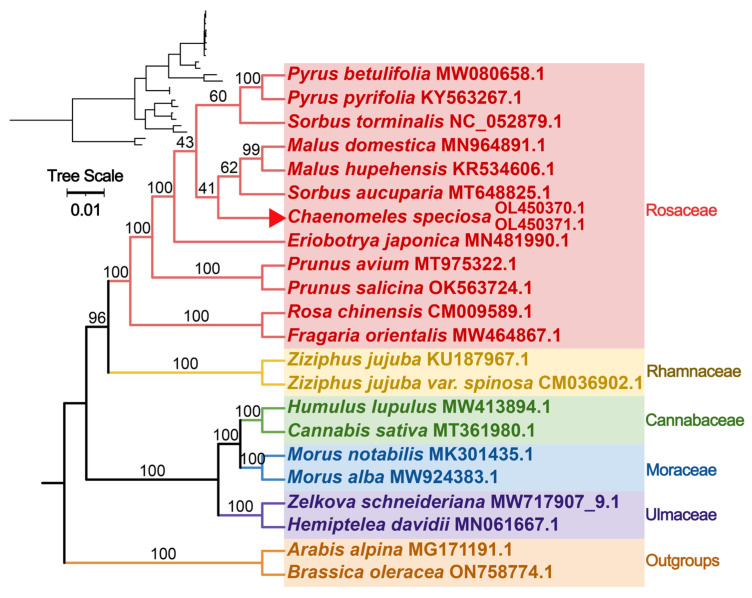
Phylogenetic tree based on 26 conserved PCGs. Two mitochondrial PCGs from *A. alpina* and *B. oleracea,* were settled as the outgroups, and the bootstrap support values were recorded at each node.

**Figure 6 genes-14-00526-f006:**
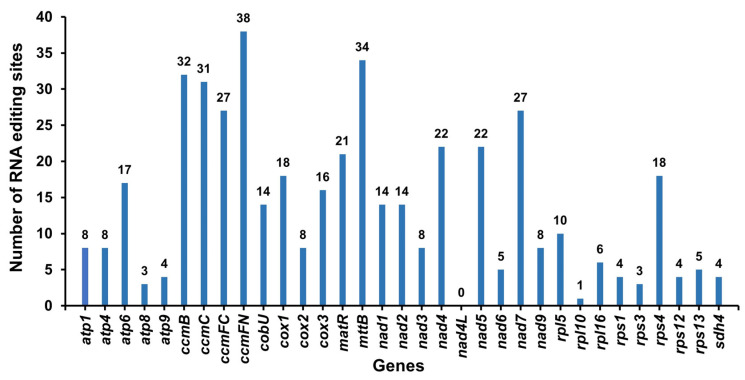
Characteristics of the RNA editing sites identified in PCGs of the *C. speciosa* mitochondrial genome. The names of the protein-coding genes are displayed on the X-axis, and the number of RNA edits of the protein-coding genes is displayed on the Y-axis.

**Figure 7 genes-14-00526-f007:**
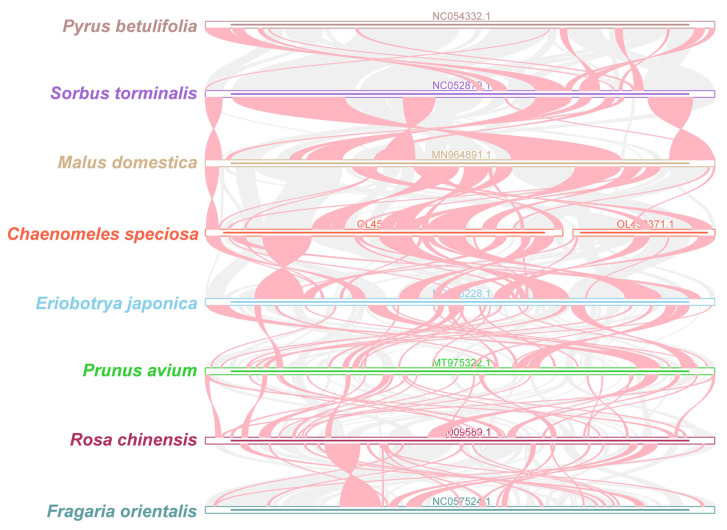
Pairwise mitochondrial genome synteny analysis of seven species of Rosaceae. The bars indicate the mitochondrial genomes, and the ribbons show the homologous sequences between adjacent species. The red areas indicate where the inversion occurred, and the gray areas indicate regions of high homology. The regions with no colinear blocks are indicated as unique in the species.

**Figure 8 genes-14-00526-f008:**
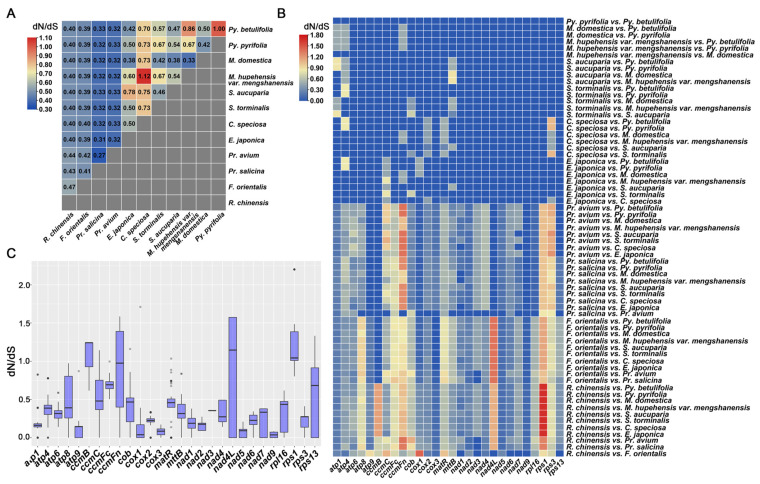
Pairwise dN/dS values in 12 species of Rosaceae: (**A**) A heatmap of the pairwise dN/dS values between each pair of sequences in the polygenic nucleotide alignment. The scale factor associated with each value is displayed at the top left of the drawing; (**B**) the pairwise dN/dS values of all PCGs in 12 species from the Rosaceae; (**C**) the boxplot of pairwise dN/dS values of all PCGs.

**Table 1 genes-14-00526-t001:** Basic information of *C. speciosa* mitochondrial genome.

NCBI Accession Number	Contigs	Type	Length	GC Content
OL450370-OL450371	Chromosome 1-2	Branched	436,464 bp	45.20%
OL450370	Chromosome 1	Circular	307,720 bp	45.15%
OL450371	Chromosome 2	Circular	128,744 bp	45.33%

**Table 2 genes-14-00526-t002:** Gene composition in the mitochondrial genome of *C. speciosa*.

Group of Genes	Name of Genes
ATP synthase	*atp*1, *atp*4, *atp*6, *atp*8, *atp*9
NADH dehydrogenase	*nad*1, *nad*2, *nad*3, *nad*4, *nad*4L, *nad*5, *nad*6, *nad*7, *nad*9
Cytochrome c biogenesis	*cob*
Ubiquinol cytochrome c reductase	*ccm*B, *ccm*C, *ccm*FC, *ccm*FN
Cytochrome c oxidase	*cox*1, *cox*2, *cox*3
Maturases	*mat*R
Transport membrane protein	*mtt*B
Large subunit of ribosome	*rpl*5, *rpl*10, *rpl*16
Small subunit of ribosome	*rps*1, *rps*3, *rps*4, *rps*12, *rps*13
Succinate dehydrogenase	*sdh*4
Ribosome RNA	*rrn*5, *rrn*18, *rrn*26
Transfer RNA	*trn*C-GCA, *trn*D-GUC, *trn*E-UUC, *trn*F-GAA (×3), *trn*G-GCC, *trn*H-GUG, *trn*K-UUU, *trn*L-CAA, *trn*M-CAU (×3), *trn*N-GUU, *trn*P-UGG, *trn*Q-UUG, *trn*S-GGA, *trn*S-UGA, *trn*T-GUA, *trn*V-GAC, *trn*W-CCA, *trn*Y-GUA

Note: The number following gene names indicates the number of copies; for example, *trn*F-GAA and *trn*M-CAU each had three copies.

**Table 3 genes-14-00526-t003:** The recombination frequency of seven repeat pairs in the *C. speciosa* mitochondrial genome.

ID	Length (bp)	Identity (%)	Position	Number of Supported Reads				Percentage (%)	
				Major 1	Major 2	Alternative 1	Alternative 2	Major Conformation	Alternative ConformatioN
R1	7887	100	chr1: 299834-307720; chr1: 107714-115600	16	18	21	20	0.4533	0.5467
R2	1235	100	chr1: 63771-65005; chr2: 1-1235	168	100	3	0	0.9889	0.0111
R4	331	100	chr1: 48854-48524; chr1: 116771-117101	161	188	1	0	0.9971	0.0029
R5	244	100	chr1: 12185-12428; chr1: 33887-34130	244	206	0	1	0.9978	0.0022
R7	163	100	chr1: 107876-107714; chr2: 121991-122153	242	237	2	0	0.9958	0.0042
R14	163	100	chr1: 299996-299834; chr2: 121991-122153	17	26	1	0	0.9773	0.0227
R15	89	96.629	chr1: 169238-169150; chr1: 207012-207100	228	146	1	0	0.9973	0.0027

**Table 4 genes-14-00526-t004:** The segments and repeats are contained in four chromosomes.

Chromosome	Order in Contigs
Contig 1	3-R12-1-R12-12-R10-10-R11-8-R4-6
Contig 2	27-R13-31-R7/R1-R7-26-R8-25-R11-24-R9-23-R6-30-R3-29-R6-28
Contig 3	22-R3-21-R5-20-R5-19-R9-18-R4-17-R8-16-R13-15
Contig 4	14-R10-13-R7

## Data Availability

Raw sequencing data were deposited at the NCBI Sequence Read Archive (SRA) under accession SRX19048963 and SRX19048964. The assembled mitogenomes were deposited in GenBank on the NCBI website (https://www.ncbi.nlm.nih.gov/, accessed on 31 January 2023) under the accession numbers OL450370.1 and OL450371.1.

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
