# Peer review of "De Novo Assembly and Comparative Analysis of the Complete Mitochondrial Genome of Chaenomeles speciosa (Sweet) Nakai Revealed the Existence of Two Structural Isomers"

_genes, 2023, doi:10.3390/genes14020526_

Round 1

Reviewer 1 Report

>Title of the manuscript is too long, need to be comprehensive and short with critical words. 

>Abstract should start from strong words not from helping words. Also need to explain the chromosome numbers and about genome size.

>Its good story but author need to address well in abstract all the section are intermingling need to improve it. 

>Figure legends need to improve by the author which are very brief. 

>Conclusions need to be more critical and straighter message for readers. 

Author Response

Dear reviewer:

Thank you for your constructive comments on my manuscript. We have carefully considered the suggestions and make some changes. We have tried our best to improve and made some revisions in the manuscript.

Please see the attachment and the revisional manuscript. 

Reviewer 2 Report

The manuscript contains appreciable information, however, the manuscript has been poorly written and needs extensive editing before ready for another round of reviewing. The major comments on each section are as follows. Please see the detailed comments/suggestions on the manuscript file attached herewith.

Abstract

Poorly written with a lot of grammatical errors and poor sentence structuring.

Introduction

Need more attention to sentence structuring and grammatical mistakes.

More literature review is needed.

The research objectives and significance of the study should be revised.

Material and methods

Many sections need extensive revisions to provide details about the analysis.

Grammatical errors are a major concern.

Do not provide results in the M&M section.

Results

Grammatical and sentence structuring errors throughout the manuscript.

Discussion

Poorly written.

Needs extensive revisions.

Hard to follow the take-home messages from most of the sections.

The full names for most of the abbreviations appear in the discussion section, which must have been introduced earlier somewhere.

Author Response

(The authors gave the same response as above.)

Round 2

Reviewer 2 Report

The manuscript has been revised properly. However, many questions and comments from the previous revision (particularly in the methodology, results and discussion sections) are not answered or ignored by the authors. Please see the file attached herewith.

The authors must understand that reviewing a manuscript is time taking process and is crucial to improve the quality of the manuscript. Therefore, the suggestions or concerns of a reviewer must be addressed properly.

Author Response

Dear reviewer:

We appreciate the comments on the manuscript of the article " De novo assembly and comparative analysis of the complete mitochondrial genome of Chaenomeles speciosa (Sweet) Nakai revealed the existence of two structural isomers". Your comments benefit us greatly in improving the quality of the manuscript. According to your advice, we amended the relevant parts of the manuscript. The point-by-point responses are attached.

Pei Cao

Best regards
